# The role of Disentanglement in Generalisation

**Milton L. Montero**[1,2], **Casimir J.H. Ludwig** [1], **Rui Ponte Costa**[2], **Gaurav Malhotra**[1] **& Jeffrey S. Bowers** [1]
1. School of Psychological Science
2. Computational Neuroscience Unit, Department of Computer Science
University of Bristol
Bristol, United Kingdom
{m.lleramontero,c.ludwig,rui.costa,gaurav.malhotra,j.bowers}@bristol.ac.uk

## Abstract

Combinatorial generalisation — the ability to understand and produce novel combinations of familiar elements — is a core capacity of human intelligence that current AI systems struggle with. Recently, it has been suggested that learning disentangled representations may help address this problem. It is claimed that such representations should be able to capture the compositional structure of the world which can then be combined to support combinatorial generalisation. In this study, we systematically tested how the degree of disentanglement affects various forms of generalisation, including two forms of combinatorial generalisation that varied in difficulty. We trained three classes of variational autoencoders (VAEs) on two datasets on an unsupervised task by excluding combinations of generative factors during training. At test time we ask the models to reconstruct the missing combinations in order to measure generalisation performance. Irrespective of the degree of disentanglement, we found that the models supported only weak combinatorial generalisation. We obtained the same outcome when we directly input perfectly disentangled representations as the latents, and when we tested a model on a more complex task that explicitly required independent generative factors to be controlled. While learning disentangled representations does improve interpretability and sample efficiency in some downstream tasks, our results suggest that they are not sufficient for supporting more difficult forms of generalisation.

## 1 Introduction

Generalisation to unseen data has been a key challenge for neural networks since the early days of connectionism, with considerable debate about whether these models can emulate the kinds of behaviours that are present in humans (McClelland et al., 1986; Fodor & Pylyshyn, 1988; Smolensky, 1987; 1988; Fodor & McLaughlin, 1990). While the modern successes of Deep Learning do indeed point to impressive gains in this regard, human level generalisation still remains elusive (Lake & Baroni, 2018; Marcus, 2018). One explanation for this is that humans encode stimuli in a compositional manner, with a small set of independent and more primitive features (e.g., separate representations of size, position, line orientation, etc.) being used to build more complex representation (e.g., a square of a given size and position). The meaning of the more complex representation comes from the meaning of it's parts. Critically, compositional representations afford the ability to recombine primitives in novel ways: if a person has learnt to recognize squares and circles in context where all squares are blue and all circles are red, they can nevertheless also recognise red squares, even though they have never seen these in the training data. This ability to perform *combinatorial generalisation* based on compositional representations is thought to be a hallmark of human level intelligence (Fodor & Pylyshyn, 1988) (See McClelland et al. (1986) for a diverging opinion).

Recently it has been proposed that generalisation in neural networks can be improved by extracting disentangled representations (Higgins et al., 2017) from data using (variational) generative models (Kingma & Welling, 2013; Rezende et al., 2014). In this view, disentangled representations capture the compositional structure of the world (Higgins et al., 2018a; Duan et al., 2020), separating the generative factors present in the stimuli into separate components of the internal representation (Higgins et al., 2017; Burgess et al., 2018). It has been argued that these representations allow

downstream models to perform better due to the structured nature of the representations (Higgins et al., 2017; 2018b) and to share information across related tasks (Bengio et al., 2014). Here we are interested in the question of whether networks can support combinatorial generalisation and extrapolation by exploiting these disentangled representations.

In this study we systematically tested whether and how disentangled representations support three forms of generalisation: two forms of combinatorial generalisation that varied in difficulty as well as extrapolation, as detailed below. We explored this issue by assessing how well models could render images when we varied (1) the image datasets (dSprites and 3DShape), (2) the models used to reconstruct these images ($\beta$-VAEs and FactorVAEs with different disentanglement pressures, and decoder models in which we dropped the encoders and directly input perfectly disentangled latents), and (3) the tasks that varied in their combinatorial requirements (image reconstruction vs. image transformation). Across all conditions we found that models only supported the simplest versions of combinatorial generalisation and the degree of disentanglement had no impact on the degree of generalisation. These findings suggest that models with entangled and disentangled representations are both generalising on the basis of overall similarity of the trained and test images (interpolation), and that combinatorial generalisation requires more than learning disentangled representations.

## 1.1 PREVIOUS WORK

Recent work on learning disentangled representations in unsupervised generative models has indeed shown some promise in improving the performance of downstream tasks (Higgins et al., 2018b; van Steenkiste et al., 2019) but this benefit is mainly related to sample efficiency rather than generalisation. Indeed, we are only aware of two studies that have considered the importance of learned disentanglement for combinatorial generalisation and they have used different network architectures and have reached opposite conclusions. Bowers et al. (2016) showed that a recurrent model of short-term memory tested on lists of words that required some degree of combinatorial generalisation (recalling a sequence of words when one or more of words at test were novel) only succeeded when it had learned highly selective (disentangled) representations ("grandmother cell" units for letters). By contrast, Chaabouni et al. (2020) found that models with disentangled representations do not confer significant improvements in generalisation over entangled ones in a language modeling setting, with both entangled and disentangled representations supporting combinatorial generalisation as long as the training set was rich enough. At the same time, they found that languages generated through compositional representations were easier to learn, suggesting this as a pressure to learn disentangled representations.

A number of recent papers have reported that VAEs can support some degree of combinatorial generalisation, but there is no clear understanding of whether and how disentangled representations played any role in supporting this performance. Esmaeili et al. (2019) showed that a model trained on the MNIST dataset could reconstruct images even when some particular combination of factors were removed during training, such as a thick number 7 or a narrow 0. The authors also showed that the model had learned disentangled representations and concluded that the disentangled representations played a role in the successful performance. However, the authors did not vary the degree of disentanglement in their models and, accordingly, it is possible that a VAE that learned entangled representations would do just as well. Similarly, Higgins et al. (2018c) have highlighted how VAEs that learn disentangled representations can support some forms of combinatorial generalisation when generating images from text. For example, their model could render a room with white walls, pink floor and blue ceiling even though it was never shown that combination in the training set. This is an impressive form of combinatorial generalisation but, as we show below, truly compositional representations should be able to support several other forms of combinatorial generalisations that were not tested in this study. Moreover, it is not clear what role disentanglement played in this successful instance of generalisation. Finally, Zhao et al. (2018) assessed VAE performance on a range of combinatorial generalisation tasks that varied in difficulty, and found that the model performed well in the simplest settings but struggled in more difficult ones. But again, they did not consider whether learning disentangled representations was relevant to generalisation performance.

Another work that has significant relation to ours is Locatello et al. (2019), who examine how hard it is to learn disentangled representations and their relation to sampling efficiency for downstream tasks. We are interested in a related, but different question: even if a model learns a disentangled representation in an intermediate layer, does this enable models to achieve combinatorial generali-

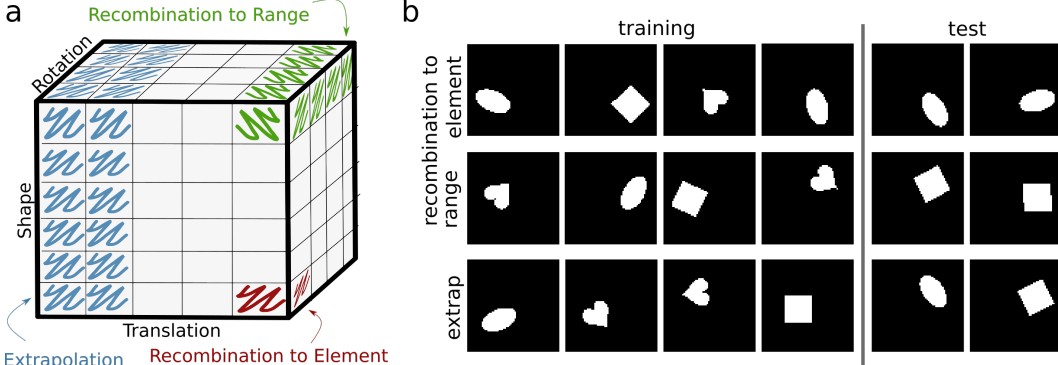

Figure 1: **Testing generalisation in image reconstruction.** (a) An illustration of different tests of combinatorial generalisation for the three-dimensional case (i.e., three generative factors). The blank cells represent combinations that the model is trained on. Coloured cells represent novel test combinations that probe different forms of generalisation: Recombination-to-Element (red), Recombination-to-Range (green) and Extrapolation (blue) – see main text for details. (b) Each row shows an example of training and test stimuli for testing a form of generalisation. In the top row, training set excludes ellipses in the bottom-right corner at less than $120°$ though they are present at the bottom-right corner at other rotations. In the middle row, training set excludes squares in the right side of the image though other shapes and rotations are present at this location and squares are seen at all other combinations of rotations and translations. In the bottom row, training set excludes all shapes on the right side of the image.

sation? So while Locatello et al. (2019) train their models on complete datasets to investigate the degree of disentanglement and sampling efficiency, we systematically exclude generative factors from training in order to test for combinatorial generalisation (see Methods and Results).

## 2 METHODS AND RESULTS

We assessed combinatorial generalisation on two different datasets. The dSprites image dataset (Matthey et al., 2017) contains 2D images in black and white that vary along five generative factors: `shape`, `scale`, `orientation`, `position-x` and `position-y` and focuses on manipulations of single objects. The 3D Shapes dataset (Burgess & Kim, 2018) contains 3D images in colour that vary along six generative factors: `floor-hue`, `wall-hue`, `object-hue`, `object-shape`, `object-scale`, `object-orientation`. In contrast to dSprites, the images are more realistic, which has been shown to aid reconstruction performance (Locatello et al., 2019). To test combinatorial generalisation, we systematically excluded some combinations of these generative factors from the training data and tested reconstruction on these unseen values. Test cases can be divided into three broad categories based on the number of combinations excluded from training.

- **Recombination-to-Element** (red squares in Figure 1): The model has never been trained on one combination of *all* of the generative factors. In dSprites, an example of this case would be excluding the combination: `[shape=ellipse, scale=1, orientation< 120°, position-x>0.5, position-y>0.5]` from the training set – i.e. the model has never seen a large ellipse at $< 120°$ in the bottom-right corner, though it has seen all other combinations.

- **Recombination-to-Range** (green squares in Figure 1): The model has never been trained on all combinations of *some* of the factors (i.e. a subset of generative factors). For example, in the 3D Shapes dataset, all combinations with `[object-hue=1, shape=sphere]` have been left out of the training set – i.e. none of the training images contain a blue sphere. This condition is more complex than Recombination-to-Element as an entire *range* of combinations `[floor-hue=0...1, wall-hue=0...1, object-hue=1, shape=sphere, scale=0...1, orientation=0...1]` have been left out (here bold text indicates the range of values excluded). When the number of generative factors is larger than three, "Recombination-to-Range" is, in fact, a set of conditions that vary in difficulty, depending upon how many generative factors have been excluded. Another example would be excluding all combinations where `[floor-hue=1, wall-hue=1, object-hue=1, shape=1, scale=1]`. Here a smaller range of

combinations `[floor-hue=1, wall-hue=1, object-hue=1, shape=1, scale=1, orientation=0...1]` have been excluded.

- **Extrapolation** (blue squares in Figure 1): This is the most challenging form of generalisation where models are tested on values of generative factors that are beyond the range of values observed in the training dataset. For example, in the dSprites dataset, all combinations where `[position-x> 0.5]` have never been seen.

Each of these conditions is interesting for different reasons. A model that learns compositional representations should be able to combine observed values of shape (ellipses), translation (bottom-right) and rotation ($0°$ to $120°$) to generalise to all unseen combination of factors. The simplest case is the recombination-to-element condition in which all combinations but one have been trained, but a model that learns entangled representations might also succeed based on its training on highly similar patterns (generalisation by interpolation). A more challenging case is recombination-to-range condition given that more combinations have been excluded, making generalisation by similarity (interpolation) more difficult. The final condition is *not* a form of combinatorial generalisation as the model cannot combine observed values of generative factors to render images. Indeed compositional representations may be inadequate for this form of generalisation.

## 2.1 IMAGE RECONSTRUCTION WITH DSPRITES DATASET

In the dSprites dataset, for testing the Recombination-to-element case, we split each range of values of a generative factor into three bins, so that we had $3 \times 3 \times 3 \times 3 \times 3$ such combinations of bins for all five generative factors. We then remove one of these $243$ combinations during training, namely those that satisfied `[shape=ellipsis, position-x >= 0.6, , position-y >= 0.6, 120° <= rotation <= 240°, scale < 0.6]`. In other words ellipses in the bottom right corner with those given rotations, which is a relatively small number of combinations that are all very similar to each other.

For the Recombination-to-range case, we tested three different variants. First, we excluded all combinations where `[shape=square, position-x>0.5]`. The model sees other shapes at those positions during training and it sees squares on the left-hand side of the screen. Thus the models experiences both generative factor values independently and has to recombine them to produce a novel image at test time. In the second case, we excluded all combinations where `[shape=square, scale>0.5]`. In the third case, we excluded all combinations where `[shape=square, rotation> 90°]`. We observed very similar results for all three cases and below we report the results for the first variant.

Finally, for the Extrapolation case, we excluded all combinations of generative factors where `[position-x > x]`. We chose a set of different values for $x$: $x \in 0.16, 0.25, 0.50, 0.75$, where $x$ is normalised in the range $[0, 1]$ (results shown in Figure 2 for $x = 0.50$). At test time the model needed to reconstruct images where translation along the x-axis, $x$, was greater than the cutoff value.

We tested three classes of models on all three types of generalisation: standard Variational Autoencoder (VAEs, Kingma & Welling (2013); Rezende et al. (2014)), $\beta$-VAE (Higgins et al., 2017; Burgess et al., 2018) with $\beta = 8$ and $\beta = 12$, FactorVAE (Kim & Mnih, 2019) with $\gamma = 20$, $\gamma = 50$ and $\gamma = 100$. The architectures are the ones found in Higgins et al. (2017), Burgess et al. (2018) and Kim & Mnih (2019) (Details in the Appendix). We used a batch size of $64$ and a learning rate of $5e-4$ for the Adam optimizer (Kingma & Ba, 2017). In each case, we simulated three seeds and we report results for runs where we obtained largest disentanglement.

As shown by Locatello et al. (2019), none of the models trained end-to-end in an unsupervised manner produce perfectly disentangled representations. Since we were interested in studying the effect of disentanglement on generalisation, we compared our results with a model where we removed the encoder and directly gave disentangled latents as inputs to the decoder. We call this model the ground-truth decoder (GT Decoder from here on). This decoder uses the same MLP architecture as the one used in Higgins et al. (2017). We tested deeper decoders with convolutions and batch norm as well, but found no benefit or a decrease in performance.

We measured the level of disentanglement using the framework introduced in Eastwood & Williams (2018). The procedure consists of using the latent representations generated for each image to predict the true generative factors using a regression model (in our case, Lasso regression; see Ap-

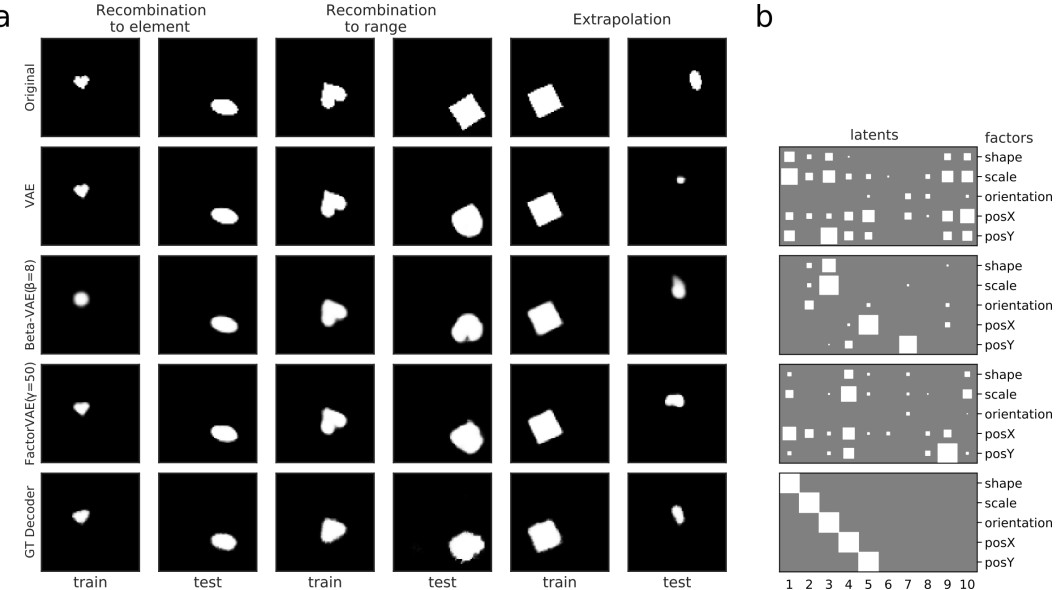

Figure 2: **Image reconstruction and disentanglement for the dSprites dataset** (a) Top row shows examples of input images and the four rows below show reconstructions by four different models. Three pairs of columns show reconstructions in training and test conditions. Left) Recombination-to-Element condition where the models did not see [shape = $ellipse$, scale $= 1$, orientation $< 120°$, position-x $> 0.5$, position-y $> 0.5$], Middle) Recombination-to-Range condition where models did not see [shape = $square$, position-x $> 0.5$], Right) Extrapolation condition where models did not see [position-x $> 0.5$] (b) Visualisation of disentanglement. In each panel, columns show latent variables and rows show the generative factors. The size of the square represents the relative importance of the latent variable for predicting the generative factor. Sparse matrices indicate higher disentanglement (Eastwood & Williams, 2018). Each disentanglement matrix corresponds to the model on that row in (a) in the Reconstruction-to-Range condition. The visualisation of the entire set of models and all conditions is shown in Appendix B

pendix A). The level of disentanglement is quantified by their 'Overall disentanglement metric', which we call D-score here.

Figure 2 shows examples of model reconstructions for each of the conditions which help assess the reconstruction success qualitatively (more examples are shown in Appendix C). A more quantitative assessment of the models can be made by examining the negative-log-likelihood of reconstructions for different conditions, plotted in Figure 3. The amount of disentanglement achieved by the models trained end-to-end varied over a broad range and was a function of model architecture and the hyper-parameter ($\beta$ and $\gamma$) values. In general, reconstruction accuracy was better for smaller values of $\beta$ both during training and testing. This has been observed before and is a known issue encountered when increasing the value of $\beta$ parameter (Hoffman & Johnson, 2016). We found that models were able to perform the Recombination-to-Element generalisation but failed in the Recombination-to-Range and Extrapolation cases. In these cases, models either showed really poor reconstruction of the critical element or substituted one of the excluded combination with a combination that had been observed during training (see reconstructions for test cases in Figure 2(a)). Moreover, the amount of generalisation did not depend on the degree of disentanglement. Indeed, the GT Decoder using perfectly disentangled representations was no better than the end-to-end models. Even though this model achieved a lower NLL score, examining the image reconstructions showed that it failed to reconstruct the essential combination excluded from the training data (see Appendix B).

The Recombination-to-Range condition shows another interesting qualitative difference between the entangled and disentangled models. All models failed to generalise, but in different ways. Entangled models tended to put a blob in the correct location, which allows them to minimise loss in pixel space over a large set of test examples. In contrast, the models with higher level of disentanglement fell back to the most similar shape (in pixel space) that they had seen at that location.

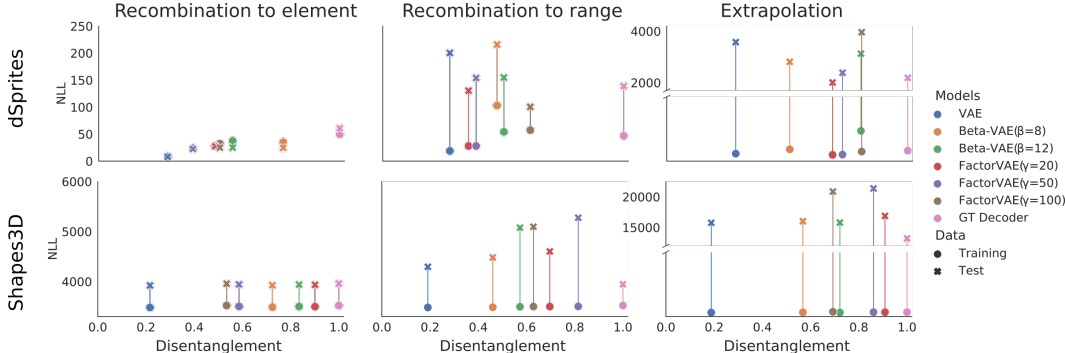

Figure 3: **Disentanglement vs reconstruction NLL**. The relation between the level of disentanglement and the performance of the model. Performance of the training data is plotted along with performance in the test (generalisation) data. Disentanglement does not provide any help in performance for the end-to-end models. The ground truth decoder (GTD) is less affected, yet it is still the case that it fails to generalize (see Figure 2 and Figure 4).

Finally, the Recombination-to-Element condition was solved by all the models, regardless of disentanglement score. In fact, the entangled models tended to achieve better reconstructions as evidenced by the disentangled models with $\beta$=12 which had a hard time reconstructing ellipses at small scales and tended to just produce a circle instead.

The second panel in Figure 2 shows the coefficients computed by the disentanglement metric for the Reconstruction-to-Range condition. The size of each square denotes the relative importance of a latent (column) in predicting the corresponding generative factor (row). The higher the disentanglement, the sparser the matrices. An examination of these matrices revealed that different models achieved a large range of disentanglement though none of the end-to-end models achieved perfect disentanglement.

## 2.2 Image Reconstruction with 3D Shapes dataset

The procedure for testing on the 3D Shapes dataset parallels the dSprites dataset above. The 3D Shapes dataset has six generative factors: `floor-hue`, `wall-hue`, `object-hue`, `object-shape`, `object-scale`, `object-orientation`. For the Recombination-to-Element condition, we excluded one combination from training: `[floor-hue > 0.5, wall-hue > 0.5, object-hue > 0.5, object-shape=cylinder, object-scale=1, object-orientation=0]`. For the Recombination-to-Range condition, we excluded all combinations where `[object-hue >= 0.5 (cyan), object-shape = oblong]` and trained all other combinations. This means that the models saw several combinations where `object-hue` was $>= 0.5$ and where `object-shape` was `oblong` but never the combination together. For the Extrapolation condition, we excluded all combinations where `[floor-hue >= 0.5]`.

We trained the same set of six end-to-end models as above as well as the GT Decoder. All end-to-end models were trained for 65 epochs (around 500000 iterations as in the original articles), while the GT Decoder was trained for 1000 epochs. Reconstructions for the training set are shown in Appendix C and clearly show that the models were able to learn the task. The results for the test conditions are shown in Figure 3 (bottom row) and some examples of typical reconstructions are shown in Figure 4. As it was the case with the dSprites dataset, we observed that the level of disentanglement varied across models, with VAE showing a low D-score and Factor-VAE showing a high Dscore. We also tested the perfectly disentangled model where a decoder learns to construct images from disentangled latents.

All models managed to reconstruct the held-out combination in the Recombination-to-element condition. However, none of the models succeeded in correctly reconstructing the held-out combinations in the Recombination-to-range or Extrapolation conditions. In both cases, we observed a large reconstruction error either due to poor overall reconstruction (Extrapolation case) or because the critical combination, `[object-hue, object-shape]` was replaced with a combination observed during training. And again, we did not see any correlation between disentanglement and the extent of combinatorial generalisation. Even though the perfectly disentangled model had a lower NLL score (see

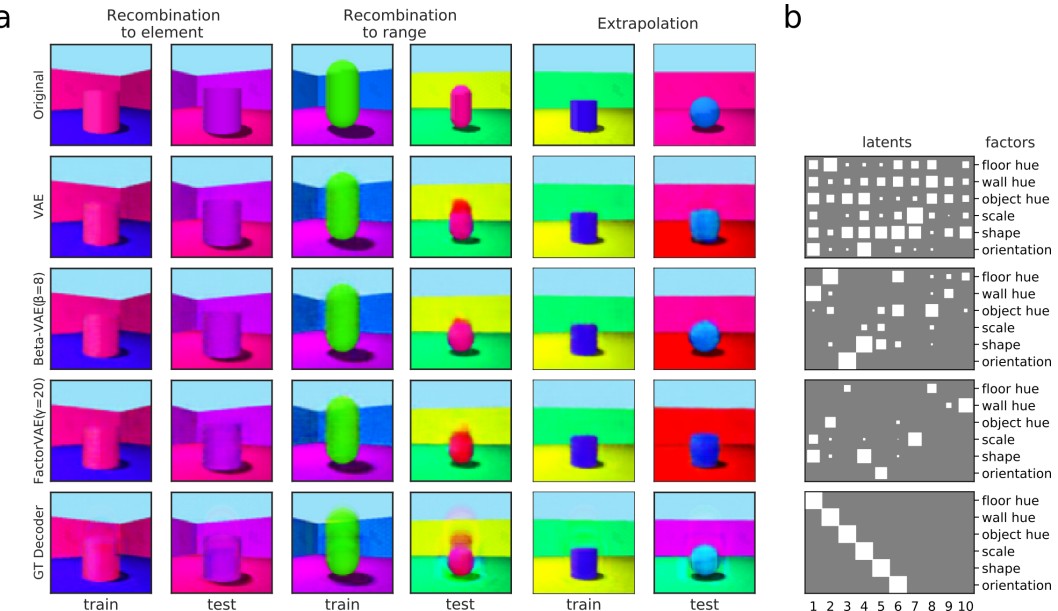

Figure 4: **Image reconstructions and disentanglement for the Shapes3D dataset**. We use the same layout as in Figure 2. a) Reconstruction examples for each of the three generalisation conditions. For the first condition, the model has not seen magenta floors with purple cylinders, yet it is able to reconstruct them properly. For the second condition, it has not seen magenta oblong shapes, yet it has seen it in other colors and it has seen magenta on other shapes. Finally, the in the third condition magenta floors have never been seen during training. b) Example Hinton diagrams of the coefficients used to compute disentanglement. Diagram in each row corresponds to the model in the same row in (a). Sparse matrices are better and the perfect one (up to permutation) is shown at the bottom.

Figure 3, bottom row), like other models it failed to reconstruct the critical `[object-hue, object-shape]` combination that was left out in the training data (see example images of reconstruction in Figure 4 and Appendix C).

## 2.3 IMAGE COMPOSITION EXPERIMENTS

The limited combinatorial generalisation in the experiments above could be because of the limitations of the task rather than the models or their internal representations. Even though the models learned disentangled representations to some extent, or were provided perfectly disentangled representations, it could be that the simple reconstruction task does not provide enough impetus for the decoder to learn how to combine these disentangled representations to enable generalisation. Therefore, in the final set of experiments we designed a variation of the standard unsupervised task that requires combining generative factors in order to solve the task using the dSprites dataset.

This new task is illustrated in Figure 5(a). The input consists of two images and an action. The goal of the task is to take the first (*reference*) image and modify it so that it matches the second (*transform*) image along the dimension specified by the action. This action is coded using a one-hot vector. This design is based on the question answering task in Santoro et al. (2017) and the compositional one in Higgins et al. (2018c). We produced training and test sets for each condition by sampling reference-transform pairs along with an action uniformly from the generative factors. We ensured that this sampling respected the experiment restriction, so that the transformed image is not outside the current set.

The standard VAE is inadequate to solve this task. Therefore, we constructed a model with the architecture shown in Figure 5(b). This model first applies an encoder to both images, obtaining low-dimensional latent representations of each image. It then combines these latent representations with the action to obtain a transformed internal representation. There are several ways in which the input representations of the two images could be combined with the action. We tried three different

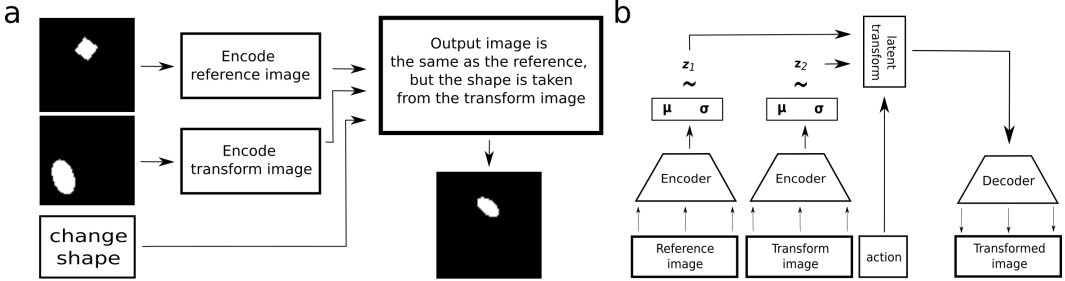

Figure 5: **Image composition task.** (a) An example of the composition task. In this case, the shape of the output must match the transform and the rest of the values must match the reference. (b) The general architecture used, based on the standard VAE. The model uses the same encoder on both images. Then a transform takes latent representation samples and combines them to produce a transformed representation. This is used to produce the transformed image.

Table 1: Model performance in the second set of experiments.

|   | Experiment | D-score | NLL (training) | NLL (testing) |
|---|---|---|---|---|
| 1 | Extrapolation | 0.82 | 31.73 | 19138.82 |
| 2 | Recomb to range | 0.71 | 50.10 | 346.10 |
| 3 | Recomb to element | 0.96 | 36.57 | 13.74 |

methods: (i) using a standard MLP, (ii) element-wise interpolation between the two representations, with the interpolation coefficients determined by the action, and (iii) concatenating each input representations with the actions and linearly combining the resultant vectors. We obtained qualitatively similar results with all three methods, but found that the method (iii) gave the best results, providing greatest accuracy of reconstruction as well as highest levels of disentanglement. Therefore, in the rest of the manuscript we describe the results obtained using this method. Once this transformed internal representation has been generated, it is decoded to obtain an output image. We use the same encoding and decoding modules as the ones used by Burgess et al. (2018). The results for this model are shown in Table 1. The model managed to solve the task. In doing so, it also comes to rely on representations with a high level of disentanglement (see Figure 6(b)) even though the $\beta$ parameter is set to 1. Models with higher values could not solve the task altogether, presumably because the constraint is too strong. However, as was the case in the previous experiment, models failed to solve the more challenging generalisation tasks.

In Figure 5 we show some examples of the model's behaviour and its internal representations. The model failed in similar ways as the disentangled models in the previous experiment, confusing shapes when presented with unseen combinations. Even the Recombination to element case showed some failures (like in the example shown in Figure 5(a)) though the models were, in general, successful in this condition, as can be inferred by comparing the negative log-likelihoods for the training and test trials for this condition in Table 1.

## 3    DISCUSSION

It is frequently assumed that disentangled representations are implicitly compositional (Higgins et al., 2018a;c). This raises the question as to whether disentangled representations support combinatorial generalisation, a key feature of compositional representations (Fodor & Pylyshyn, 1988). However, we found no evidence for this. Indeed representations that varied from highly entangled to perfectly disentangled were equally successful at recombination-to-element generalisation, and both failed on recombination-to-range and extrapolation. This was the case even when we trained a VAE on an explicitly combinatorial task that led models to learn highly disentangled representations that were no better at generalisation.

Our findings might seem to contradict previous reports showing success in combinatorial generalisation tasks. In Eslami et al. (2018), some success was reported when rendering novel 3D shapes

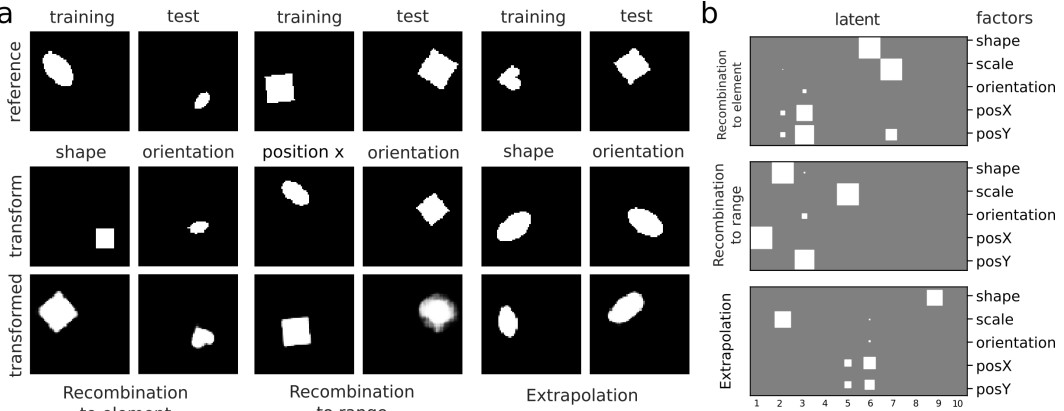

Figure 6: **Image composition generalisation results** (a) Each column shows an example trial which consists of a *reference* image, an *action*, a *transform* image and the *transformed* (output) image. We show examples of both training and test trials. Each of the training trials results in the correct (expected) transformed image, while each of the test trials shows a failure. (b) Visualisation of the degree of disentanglement for three conditions by the model. The sparse representation reflects the high level of disentanglement achieved by these models in this task.

with colours that had been previously seen on other shapes. And in Higgins et al. (2018c) it was reported that using a disentangled representation allowed the model to recombine observed shapes and colours in novel ways. However, it is not clear what sorts of combinatorial generalisation was tested. For example, consider the SCAN model (Higgins et al., 2018c) that could render a room with `[white suitcase, blue walls, magenta floor]`, even though it was never shown this combination during training (see Figure 4 in Higgins et al. (2018c)). But, unlike our training set, it is not clear what exactly was excluded while training this model, and they may have been testing generalisation in a condition similar to our Recombination-to-element condition. Our finding that generalisation was limited to the Recombination-to-element condition suggests that models are simply generalising on the basis of overall similarity (interpolation) rather than exploiting disentangled representations in order to support a more powerful form of compositional generalisation described by Fodor & Pylyshyn (1988).

This raises the question as to why disentangled representations are not more effective in supporting combinatorial generalisation. One possibility is that disentangled representations are necessary but not sufficient to support the principle of compositionality. On this view, a model must also include a mechanism for binding these representations in a way that maintains their independence. This point has previously been made in the context of connectionist representations by Hummel in Hummel (2000). Another possibility is that a model may be able to perform combinatorial generalisation without needing disentangled or indeed compositional representations if the training environment is rich enough (Chaabouni et al., 2020; Hill et al., 2020; Lampinen & McClelland, 2020).

An important goal for future research is to develop networks that support the more difficult forms of combinatorial generalisation and extrapolation. In fact there is already an active range of research in this direction that include networks with specialized modules (Santoro et al., 2017), mechanisms (Mitchell & Bowers, 2020; Hummel & Biederman, 1992), structured representations (Higgins et al., 2018c; Watters et al., 2019), or learning objectives (Vankov & Bowers, 2020) that may show greater success. It will be interesting to see how these and other approaches fare in the more difficult generalisation settings we have identified here, and the role of disentanglement in any solutions.

## ACKNOWLEDGEMENTS

We would like to thank Chris Summerfield, Irina Higgins, Ben Evans and Jeff Mitchell for useful discussions and feedback during the development of this research.

This research was supported by a ERC Advanced Grant (Generalization in Mind and Machine, #741134).

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

# A    MODELS AND TRAINING

For our experiments on the standard unsupervised task we used two different VAE architectures. The first one is the same found in Higgins et al. (2017) and uses a 2-layer MLP as an encoder with 1200 units and ReLU non-linearity. The decoder is a 3-layer with the same number of units and the Tanh non-linearity. The second architecture is the one found in Burgess et al. (2018) and consists of a 3-layer CNN with $32 \times 4 \times 2 \times 1$ convolutions and max pooling, followed by a 2-layer MLP with 256 units in each layer. The decoder is defined to be the transpose of this architecture. ReLU non-linearity where applied after each layer of the CNN and the MLP for both the encoder and the decoder. Both models used a Gaussian stochastic layer with 10 units as in the original papers.

We also tested two variants of this last architecture, one found in Mathieu et al. (2019) which changes the shape of the convolution and another with batch normalisation. Neither variant exhibited any improvements to disentanglement or reconstruction on the full dSprite data and so were not included in the rest of the experiments.

For the image composition task we used same as in Burgess et al. (2018) that we described above. The latent transformation layer was parameterized as:

$$h_{transformed} = W_r cat[z_r; action] + W_t cat[z_t; action]$$

where $z_r$ and $z_t$ are the samples from the stochastic layer for reference and transform image, $cat$ is the concatenation operation performed along the column dimension. The output is another 10-dimensional vector with the transformed latent code.

Alternatively we also tried a 3 layer MLP with 100 hidden units, but saw no benefit in performance and decreased disentanglement when trainined on the full dataset.

Training on the unsupervised tasks ran for 100 epochs for dSprites and 65 epochs for Shapes3D, even though models converged before the end. The learning rate was fixed at $1e - 4$ and the batch size at 64. $\beta$ values used were $1, 4, 8, 12, 16$ on the full dSprite dataset. $\beta = 4$ and $\beta = 16$ where not included in the rest of the experiments since the former offered very little disentanglement and the latter very large reconstruction error. For the FactorVAE we used $\gamma = 20, 50, 100$ throughout. In the composition task the models where trained for 100 epochs with $\beta = 1$. Using $\beta$ higher than 1 interfered with the model's ability to solve the task so they where not used.

For the ground-truth decoders (GT Decoder) we used the same MLP decoder of Higgins et al. (2017) mentioned above. Using deeper decoders with convolutions with/without batch norm after each layer was also tested, but did not provide significan benefits and also decreased the performance on some of the conditions.

All the models where implemented in PyTorch (Paszke et al., 2019) and the experiments where performed using the Ignite and Sacred frameworks (V. Fomin & Tejani, 2020; Klaus Greff et al., 2017).

To measure disentanglement we used the framework proposed by Eastwood & Williams (2018) with a slight modification. The approach consists of predicting each generative factor value, given the latent representations of the training images using a non-linear model. In our case we used the LassoCV regression found in the Sklearn library (Pedregosa et al., 2011) with an $\alpha$ coefficient of 0.01 and 5 cross-validation partitions. Deviating from the original proposal, we do not normalize the inputs to the regression model since we found that this tends to give a lot of weight to dead units (when measured by their KL divergence). This is likely due to the model "killing" these units during training after they start with a high KL value, which might not completely erase the information they carry about a given generative factor.

Working code for running these experiments and analyses can be downloaded at `https://github.com/mmrl/disent-and-gen`.

# B    EXTRA PLOTS FOR DSPRITES DATASET

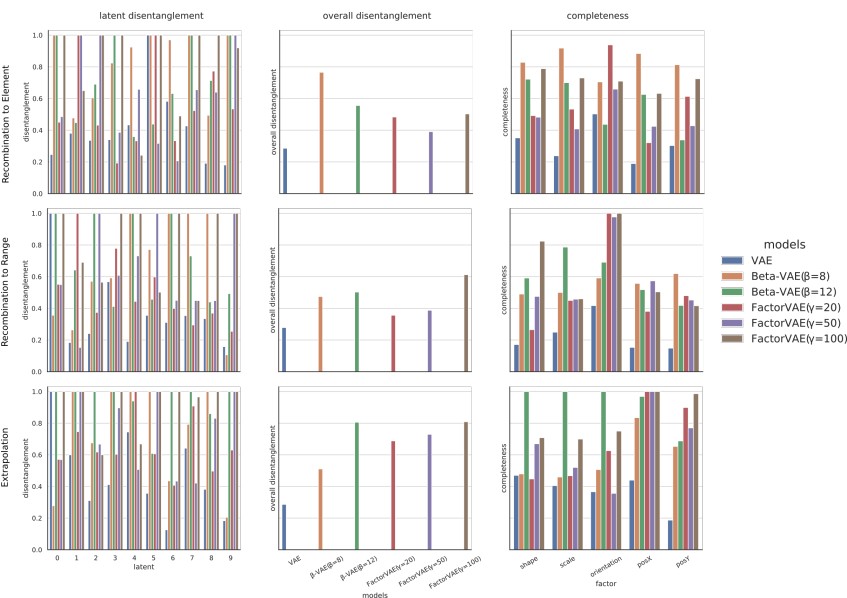

Figure 7: **Disentanglement scores for dSprites**. The disentanglement analysis results for the dSprites dataset. The scores for each of the metrics evaluated by the DCI framework: disentanglement (left), overall disentanglement (middle) and completeness (right) for each of the conditions.

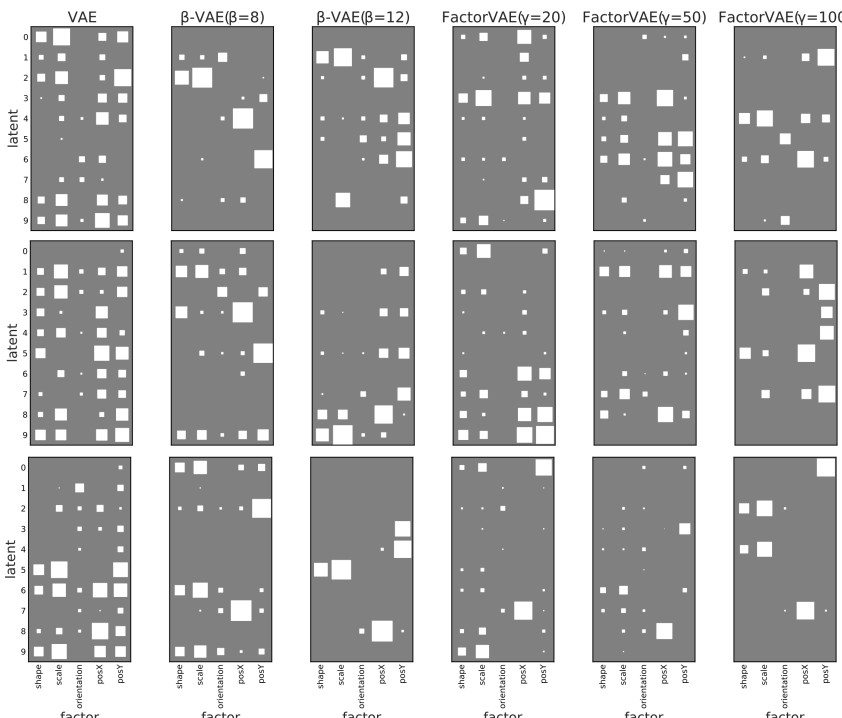

Figure 8: **Hinton diagrams for dSprites dataset**. The matrices of coefficients computed by the framework plotted as Hinton diagrams. These are used to obtain the quantitative scores in the panel above. They offer a qualitative view of how the model is disentangling. On the left is how perfect disentanglement looks in this framework.

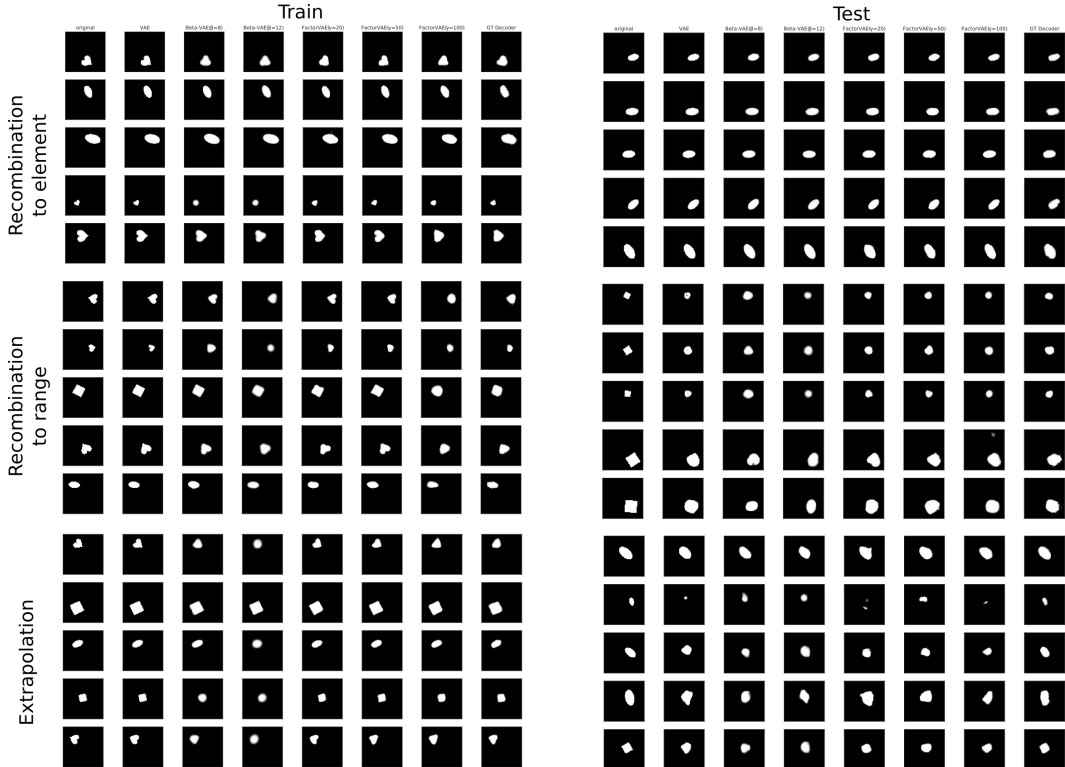

Figure 9: **Reconstructions for the dSprites dataset**. For each condition and model, these are some reconstruction examples for both training and testing. There is general success and failure in the Recombination to Element (top) and Extrapolation (bottom) conditions, respectively. For this last condition, the models seem to reproduce the closest instance they have seen, which tranlated to the middle of the image. For the Recombination to range (middle), the models tend to resort to generating a blob at the right location, minimising their pixel-level error.

## C  EXTRA PLOTS FOR THE 3D SHAPES DATASET

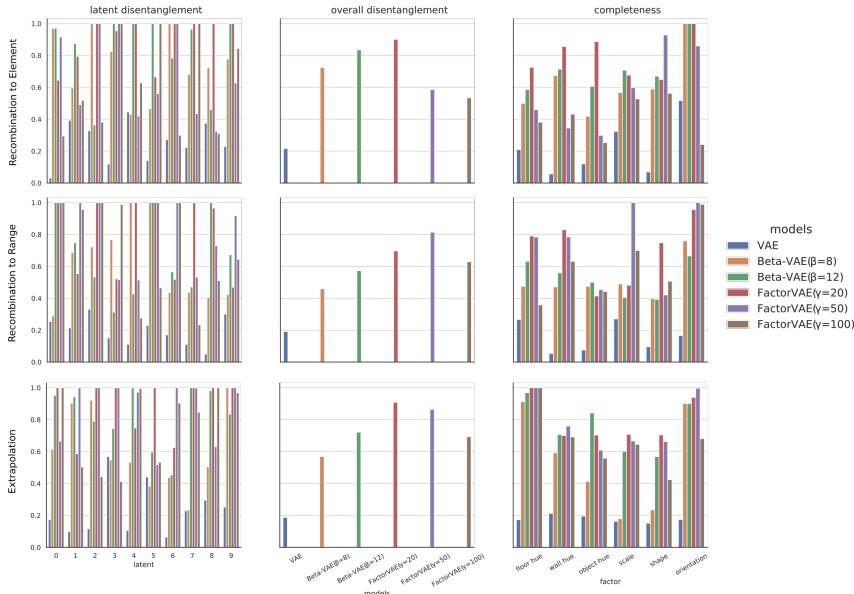

Figure 10: **Disentanglement analysis for Shapes3D**. The disentanglement analysis results for the 3D Shapes dataset. The scores for each of the metrics evaluated by the DCI framework (Eastwood & Williams, 2018): disentanglement (left), overall disentanglement (middle) and completeness (right) for each of the conditions.

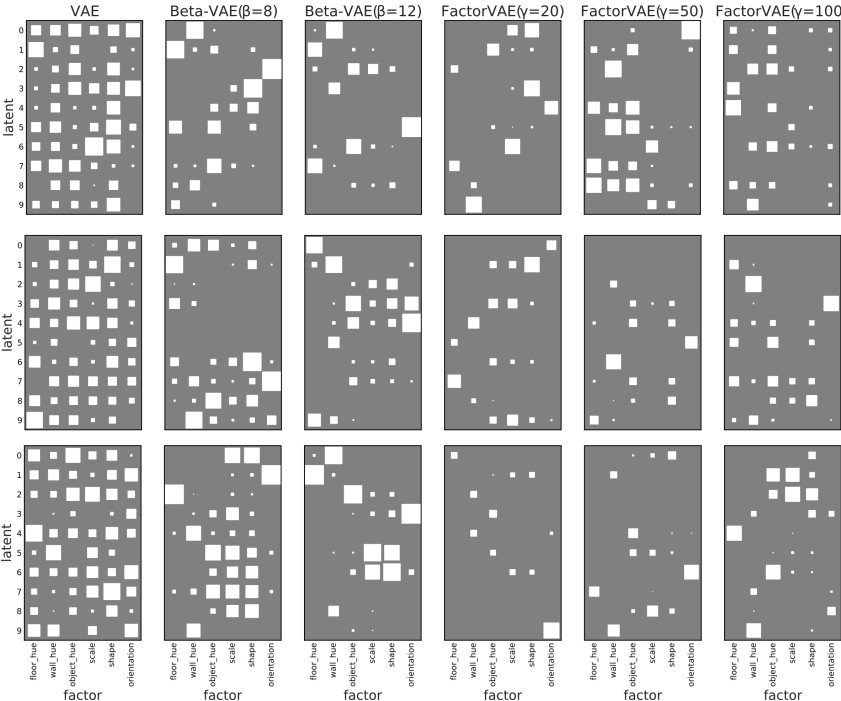

Figure 11: **Hinton diagrams for 3DShapes dataset**. The matrices of coefficients computed by the framework plotted as Hinton Diagrams. As discussed in the main text, these matrices offer a qualitative view of how the model is disentangling. In general, sparse matrices indicate higher disentanglement. It is clear from these diagrams that the degree of disentanglement varies over a broad range for the tested models.

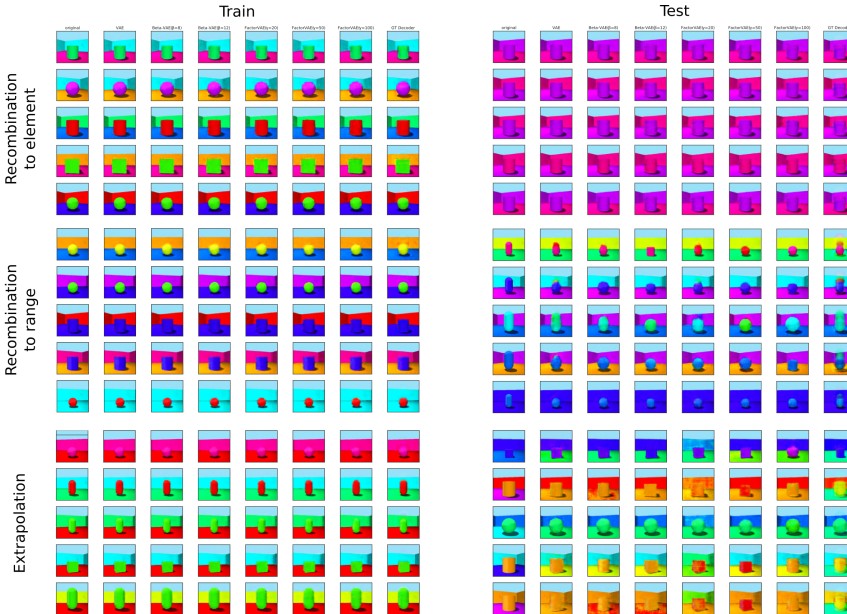

Figure 12: **Reconstructions for the Shapes3D dataset**. For each condition and model, these are some reconstruction examples for both training (left) and testing (right). In each case, the input image is shown in the left-most column and each subsequent column shows reconstruction by a different model. The test images always show a combination that was left out during training. All training images are successfully reproduced. However, reconstruction for test images only succeeds consistently for the Recombination-to-Element condition (top). All reconstructions fail in the Extrapolation condition (bottom) while most of them fail for the Recombination-to-Range condition (middle). There are occasional instances in Recombination-to-Range condition that seem to correctly reconstruct the input image. This seems to happen when the novel combination for color and shape is closest to the ones the model has experienced during training. For example, models are better when the oblong shape is paired with cyan (which is close to green, which it has seen) and worse on magenta.

