# OpenReview forum: "The role of Disentanglement in Generalisation"
_ICLR.cc/2021/Conference — ICLR 2021 Poster_

### Official Review · AnonReviewer4 · 2020-10-13
**A good start, but restricted experiments limit the conclusions, and the discussion could be refined**

**Rating:** 8
**Confidence:** 5

**Review:**

Post-revision update
------------------------

Thanks to the authors, I think that the revision provided by the authors makes the paper substantially stronger. The inclusion of the more complex Shapes3D dataset substantially improves the experiments, and I think the discussion has improved. I have revised my rating to a clear accept in accordance.

Original Review
----------------------

This paper evaluates the role of disentanglement in generalization. The authors begin by articulating a useful distinction between different kinds of generalization by interpolation, recombination or extrapolation. They then train VAEs (and variations thereof) on a controlled, synthetic dataset, using two different training paradigms. They show that the models only generalize well to one of the most elementary types (recombination to element), but do not extrapolate well to the more difficult kinds of generalization. They also show that disentanglement does not seem to correlate with better generalization.

I find this paper to be marginally above the acceptance threshold, but it has room for improvement (see below).

Strengths:

* Generalization and the role of compositionality and disentanglement therein are very important issues.

* I like the articulation of the different kinds of generalization, and the clear illustration thereof.

Areas for improvement:

* The relationship of these results to Locatello et al. (2019) would be worth discussing further. They also showed that disentanglement did not necessarily lead to better generalization, with a wider range of experiments (though therefore perhaps less deep in evaluating different types of generalization).

* The experiments are very narrow. The paper uses a single dataset (though with two different tasks), and it is very toy. As noted by Locatello et al. (2019), the inferences drawn from a single dataset may be very biased. While simple synthetic datasets can be useful for allowing more carefully controlled experiments, it would be useful to explore the same experiment
s using different datasets with different features (e.g. color and texture). It would also be useful to understand generaliza
tion in richer, more realistic datasets (see below).

* It would be useful to show some of the major claims in a less opaque way. For example, to show the (non)relationship between disentanglement and generalization, the authors could make a plot with D-score on the x-axis, and generalization performance on the y-axis (with different plot panels for the different types of generalization, perhaps).

* One reason that exploring other datasets is important is that the particular inductive biases of the models may facilitate generalization along certain feature dimensions. For example, the fact that convolutions are generally (relatively) translation-invariant but not rotation-invariant mean that the model might more easily extrapolate to unseen translations. In order to draw broad conclusions, it would be useful to both explore more diverse datasets and quantitatively analyze in more detail t
he generalization along different dimensions.

* Increasing realism can produce qualitative improvements in compositional generalization in some settings. For example, Hill et al. (2020) showed that e.g. generalization was better in a 3D setting than 2D setting, and that an RL agent showed 100% compositional generalization in a setting where a classifier only showed ~80%, for instance (although this generalization was recombination, not extrapolation). Thus, the poverty of the stimuli may alter the paper's conclusions. Even if your study is useful, it's worth discussing this limitation more explicitly.

* The paper raises the question of why disentangled representations are not more effective in supporting compositional generalization, but it's worth asking why we assume that they would. Disentanglement $\neq$ compositional representations $\neq$ systematic generalization. Fodor & Pylyshyn suggest that compositional representations are necessary for systematic generalization, but they don't certainly don't provide an empirical definition of how to evaluate compositionality of representations. Indeed, it's hard to define such a notion: "The question of whether a model [generalizes] according to compositional task structure is distinct from the question of whether the model’s representations exhibit compositional structure. Because the mapping from [...] representations to behavior is highly non-linear, it is difficult to craft a definition of compositional representations that is either necessary or sufficient for generalization" (Lampinen & McClelland, 2020). Your results, along with those of Locatello et al (2019) and others, lend support for this argument. Disentanglement in some middle layer of the network does not seem to show a causal role in generalization, presumably in part because the processes intervening between that representation and the output are nonlinear, because that nonlinear decoder is also capable of failing for some combinations of latent representations even if the representations themselves are compositional, and/or because disentanglement is not a sufficient notion of compositionality. Given these difficulties, I think you could potentially extrapolate further than you do, to ask whether we should be worrying about representations at all, rather than just evaluating (and improving) behavioral performance on the different types of generalization you articulate.

* However, even empirical evaluation is challenging in more naturalistic settings. The notion of "disentanglement" or "composition" may be harder to define in realistic datasets. It's not clear what the appropriate decomposition of a complex naturalistic image is — objects are a natural place to start, which is why Higgins and others have focused on this type of decomposition. But what counts as disentangled in a visual scene of e.g. a forest? Is each leaf an object that must be disentangled in its own right? Should the color of each piece of bark on each tree be represented by its own dimension, since *in principle* it could vary independently? This seems unreasonable, which is perhaps why disentanglement is usually demonstrated on very simplistic datasets. Yet a human can of course *attend* to any particular aspect of the scene to disentangle that dimension as needed. In the real world, human-like performance might require the ability to construct *new decompositions on the fly,* because the appropriate decompositions may change as the task or data shifts. That is, the idea of seeking a priori disentanglement with respect to fixed dimensions might not be the right way to go about achieving human-like generalization, especially if we want that generalization to extrapolate to new data and new tasks. (C.f. Lampinen & McClelland, 2020 for some other related discussion.)

* It also seems likely that the processes that allow humans to exhibit strong generalization may require extended or additional processing, rather than a single feed-forward pass as in a VAE. This would be necessary to allow the sort of attentive disentanglement described in the previous point. For example, the Stroop effect in cognition seems to me to illustrate feature entanglement, which requires higher level control processes to resolve the appropriate response. The paper does discuss the idea that other mechanisms or architectures might be involved in the discussion, but it seems it bears more elaboration, especially w.r.t. the above points about the definability of disentanglement with real world data.



References
-----------

Hill, Felix, et al. "Environmental drivers of systematicity and generalization in a situated agent." International Conference on Learning Representations. 2020.

Lampinen, Andrew K., and James L. McClelland. "Transforming task representations to allow deep learning models to perform novel tasks." arXiv preprint arXiv:2005.04318 2020.

Locatello, Francesco, et al. "Challenging common assumptions in the unsupervised learning of disentangled representations." international conference on machine learning. 2019.

---

> ### Author Response · Authors · 2020-11-24
> **Response to AnonReviewer4: Part I**
>
> Thank you for your detailed comments and feedback. We respond to each point below:
>
> Response to “The relationship of these results... types of generalisation”: We have now modified Section 1.1, where we discuss the Locatello et al. (2019) paper and their more recent paper (van Steenkiste et al.). These papers find that (a) unsupervised methods cannot always identify disentangled models and (b) disentangled representations do not necessarily lead to decreased sample complexity for downstream tasks. Here, we were focused on the specific case of combinatorial generalisation and its relation with disentanglement. Our new simulation makes this point in the most emphatic manner: even when latent representations are completely disentangled, it does not guarantee combinatorial generalisation. To test this, we left out various combinations from the training dataset, which is quite different to the tests conducted by Locatello et al. (2019). But we agree that there is a close connection between the two studies, and we have now made this clear in the manuscript.
>
>
>
> Response to “The experiments are very narrow...”:   This was a concern shared by many reviewers. Therefore, we have now run a new set of experiments expanding and replicating our findings. Please see our general response to all reviewers, where we have discussed the new experiments and findings.
>
>
>
> Response to “It would be useful to show... , perhaps)”: Thank you – this is really useful feedback. We have now done exactly this, replacing Table 1 with Figure 3. If generalisation improves as a result of disentanglement, NLL should decrease as disentanglement increases. This is clearly not the case for any of our test conditions. The only exception to this is the NLL score for the perfectly disentangled (decoder) model in the Recombination-to-Range condition. But even here, the lower NLL value is misleading as the models reconstruct crucial elements of the image (the combination of left-out generative factors) incorrectly. This can be seen by looking at the example reconstructions in Figures 2, 3 and in the Appendix Figures 8 & 10.
>
>
>
> Response to “One reason... different dimensions”: We have now added the 3D Shapes dataset and explored different forms of generalisation in these datasets.  For example, in the dSprites dataset we explore extrapolation using translation and in the 3D dataset we explored extrapolation along the color dimension of the floor.
>
>
>
> Response to “Increasing realism... more explicitly”: We thank the reviewer for pointing out the Hill et al. (2020) paper.  The paper makes the important point that generalisation is better (albeit still limited) when the training environment is richer (effectively making the generalisation less difficult). This is somewhat akin to our finding that the models were able to generalise in the Recombination-to-element condition as lesser combinations were left out. The key difference between Hill et al and our paper is that we were interested in the role of disentanglement in generalisation, which is not something that was explored there. We now cite this paper (Discussion) and highlight the need to explore more tasks, more training environments, more architectures, and the role of disentanglement in order to achieve more human-like generalisation.
>
>
>
> Response to “The paper raises the question... generalisation you articulate”: We agree with the reviewer’s comment that “Disentanglement ≠ compositional representations ≠ systematic generalization”.  We now explicitly make this point in the General Discussion. As we mentioned above, we have now explicitly added an experiment where we start with perfectly disentangled representations and train the decoder to reconstruct based on these representations. As predicted by the reviewer, this model fails in the same conditions as the end-to-end models. However, we do not want to make the strong conclusion that we should focus on performance at the expense of worrying about representations.  For example, we cite Hummel (2000) who argues that disentangled (localist) representations are useful (but not sufficient) for implementing symbolic computations needed for broader generalisation.  We are not committed to this view either, but we do think that considering different architectures and different types of inductive biases that impact on representations is something that needs to be explored alongside focusing on performance in richer training environments.

---

> ### Author Response · Authors · 2020-11-24
> **Response to AnonReviewer4: Part II**
>
> Response to “However, even empirical evaluation... related discussion)”: Thank you for pointing out the Lampinen & McClelland (2020) paper. This is an interesting perspective that we hadn’t come across before. We agree, disentanglement may not be the solution to some (or indeed many) tasks.  We also agree that it is more difficult to know what factors could be disentangled in more naturalistic settings, and indeed, that is why we focused on the dSprites and 3DImage datasets as a starting point.
>
>
>
> Response to “It also seems likely... real world data”: We are not sure we agree with the reviewer that the Stroop effect reflects the impact of entangled representations – colour and shape might be disentangled (and indeed there is good evidence for this from Garner interference) with the interference the product of a response conflict.  That is, colour and shape representations may be processed separately using disentangled representations, and the delay in responding reflects the conflicting outputs of these two processes (e.g., colour channel outputting “blue” and word channel outputting “red”).  But we do agree that more complicated forms of generalisation may need more than a feed-forward pass.  Our findings highlight the limitations of disentangled presentations in this context, and we agree with the reviewer that more complex environments, tasks, and architectures are likely needed.

---

> > ### Comment · AnonReviewer4 · 2020-11-24
> > **Thanks, I think the paper is substantially improved**
> >
> > Thanks to the authors for the response and for the additional experiments, I think the paper has improved significantly and have updated my rating accordingly.

---

### Official Review · AnonReviewer1 · 2020-10-27
**Interesting study, but needs more experiments**

**Rating:** 6
**Confidence:** 4

**Review:**

Summary:
This paper studies the performance of models producing disentangled representations in the downstream task of combinatorial generalization. The experiments suggest that models producing disentangled representations do not generalize well enough.

Pros:
- The paper is well-written and easy to follow.
- The authors propose four novel benchmarks to systematically study the ability of a model to generalize.

Concerns:
- The key concern is that the paper does not present enough experiments to support the authors' claims. The study was conducted only for one dataset; I would suggest to include several other datasets in your study, e.g., MPI 3D, Shapes 3D, Cars 3D datasets. Also, the results would be stronger if the paper presented the assessment of other disentanglement specific metrics; see, for example, MIG [1], Modularity [2], etc.

Comments/questions:
- The combinatorial generalization task looks similar to the abstract reasoning task; it was shown that disentangled representations help in this downstream task [3]. How do you think, why it does not hold as well for combinatorial generalization?
- Perhaps it would be interesting to vary random seeds in addition to $\beta$ values; it was shown in Locatello [4] that random seeds sometimes have a stronger influence on disentanglement scores than model hyperparameters.

Minor comments:
- In some places, you write "generalization", in other -- "generalisation".

UPD: The authors addressed my concerns and added additional experiments. The paper is improved, therefore, I increase the rating.

References:

[1] Ricky TQ Chen, Xuechen Li, Roger B Grosse, and David K Duvenaud. Isolating sources of disen- tanglement in variational autoencoders. In Advances in Neural Information Processing Systems, pp. 2610–2620, 2018.

[2] Karl Ridgeway and Michael C Mozer. Learning deep disentangled embeddings with the f-statistic loss. In Advances in Neural Information Processing Systems, pp. 185–194, 2018.

[3] van Steenkiste, Sjoerd, et al. "Are Disentangled Representations Helpful for Abstract Visual Reasoning?." Advances in Neural Information Processing Systems. 2019.

[4] Locatello, Francesco, et al. "Challenging common assumptions in the unsupervised learning of disentangled representations." international conference on machine learning. 2019.

---

> ### Author Response · Authors · 2020-11-24
> **Response to AnonReviewer1**
>
> The reviewers key concern was that we did not carry out enough experiments to rigorously test our hypotheses. This is a concern that was shared by other reviewers. Therefore, we have now run a set of new experiments testing the robustness of our findings. Please see our general comments to all reviewers where we describe these new experiments and findings.
>
>
>
> Response to “The combinatorial generalization task... generalization?”: The van Steenkiste et al. study shows that learning disentangled latent representations for images led to faster learning in a visual reasoning tasks that employed those images. But it is important to note that van Steenkiste et al. did not exclude specific training conditions in order to test combinatorial generalisation or extrapolation in the visual reasoning task. We do not dispute that learning disentangled representations may be useful for downstream tasks as it may improve sampling efficiency. However, performing combinatorial generalisation requires an ability to syntactically combine these disentangled representations to form new combinations, which current models lack. In fact, in another visual reasoning task Barrett et al. (2018) excluded specific training conditions and found that performance in the most difficult combinatorial generalisation conditions was “strikingly poor”, even after they modified the model and training conditions to improve performance.  The van Steenkiste et al. finding that disentangled representations improve learning in downstream tasks is important, but it does not provide evidence that these representations improve more difficult forms of generalisation.
>
> Barrett, D. G., Hill, F., Santoro, A., Morcos, A. S., & Lillicrap, T. (2018). Measuring abstract reasoning in neural networks. arXiv preprint arXiv:1807.04225.
>
>
>
> Response to “Perhaps it would be interesting... model hyperparameters”: We did indeed vary random seeds in addition to beta values. The results we show in the manuscript are for the seeds that were able to accomplish highest performance and largest disentanglement. We have now made this clear in the manuscript (pg 4). It should also be noted that we have now run a simulation where perfectly disentangled representations used in our decoder model supported limited generalisation. This suggests that the there are no β values that would lead to higher levels of disentanglement that would in turn support better generalisation.
>
>
>
> Response to “In some place... generalisation”: We have now consistently adopted the English spelling, although in the search terms we include both so that paper is easier to find.

---

### Official Review · AnonReviewer2 · 2020-10-29
**Interesting and relevant analysis, but the conclusions aren't clear enough yet**

**Rating:** 7
**Confidence:** 4

**Review:**


Summary
---
A large body of work creates disentangled representations to improve
combinatorial generalization. This paper distinguishes between 4 types of
generalization and shows that existing unsupervised disentanglement approaches
generalize worse to some and better to others.

(introduction)
There are 3 types of combinatorial generalization. Each requires a learner to
generalize to a set of instances where 0, 1, or 2 dimensions have been completely held out.
Previous work has not distinguished between these kinds of generalization when
testing how disentangled representations generalize. This work does that to
understand the relationship between disentanglement and combinatorial
generalization in a more fine grained manner.

(approach)
Throughout this paper, beta-VAE and a recent variant are trained with varrying levels of
disentanglement (controlled by beta) to reconstruct d-sprites images.
These images contain simple shapes and are generated using 5 ground truth
latent factors. The ground truth latent factors allow disentanglement to be
measured (using Eastwood and Williams 2018), essentially by checking whether
the ground truth latent factors are linearly separable in the learned latent space.

(experiment - plain reconstruction)
* reconstruction error differs for different types of combinatorial generalization (holding out fewer dimensions is easier)
* reconstruction error is not highly correlated with disentanglement

(experiment - compositional reconstruction)
Instead of reconstructing the input, a version of the input with one attribute changed is generated.
* generation error differs for different types of combinatorial generalization (holding out fewer dimensions is easier)

(conclusion)
Usually disentanglement is encouraged to achieve combinatorial generalization, but this paper presents a simple experiment where it doesn't do that.



Strengths
---

The central claim of the paper may help clarify the disentanglement literature.

It seems very useful to taxonomize generalization in this way.

The writing and motivation is generally very clear. The figures are easy to understand and help demonstrate the narrative.

This paper aims to characterize an existing line of work in detail rather than proposing a new approach/dataset/etc. I like work of this nature and would like to see more like it.



Weaknesses
---


1. The relationship between disentanglement and generalization is clearly or quantitatively demonstrated:

The most interesting claim in this paper is that disentanglement is not necessarily correlated with combinatorial generalization, but this claim is not clearly supported by the data.

* The main support comes from table 1. Here higher D-score does not necessarily mean lower test NLL. This observation should be made quantitative, probably just by measuring correllation between D-score and test NLL.

* Table 2 seems to contradict this claim. In that case higher D-score does mean lower test NLL.


2. The taxonomy of generalization is a bit too specific to be useful and a bit incoherent:

The difference between "Interpolation" and "Recombination to element" generalization
is not clear to me. Each of the purple and red cubes in figure 1a represents
a combinations of rotation, shape, and translation factors.
It may be that it makes a difference when some dimensions are categorial
and others are continuous, as in the Interpolation example, but this doesn't
seem to really solve the factor because continuous latent variables
are still latent variables. I see some vague intuition behind this distinction,
but the paper does correctly identify the precise distinction.

Furthermore, this taxonomy of generalization seems limited to me.
It seems like "Recombination to element", "Recombination to range", and "Extrapolation"
just hold out a different number of dimensions (e.g., "none", "rotation", and "shape and rotation", respectively).
This begs the question of what happens when there are 4 generative dimensions?
Is generalization when 3 of those are held out also called "Extrapolation"?

I think more work needs to be done to create a taxonomy which precisely and clearly generalizes
to N latent factors and creates a more coherent distinction between combinatorial and
non-combinatorial generalization.
However, I think it's possible to create a better taxonomy and that it
will probably be very useful to do so.


3. The paper should test the idea more thoroughly, on more datasets and on more disentanglement approaches. For example, it could include other datasets or tasks with different ground truth factors of variation (e.g., 3D chairs [1]). It could also include more disentanglement approaches like [2].


[1]: M. Aubry, D. Maturana, A. Efros, B. Russell, and J. Sivic. Seeing 3d chairs: exemplar part-based 2d-3d
alignment using a large dataset of cad models. In CVPR, 2014.
[2]: Esmaeili, B. et al. “Structured Disentangled Representations.” AISTATS (2019).





Comments / Suggestions
---

Describe the disentanglement metric in more detail. From the beginning disentanglement is treated differently from combinatorial generalization. It's not immediately clear what disentanglement is that makes it different and why that's interesting to study. For example, initially one might think that beta-VAE is inherently disentangled.

Can this taxonomy of generalization be generalized to continuous domains? For example, can it be generalized to any (typically continuous) hidden layer a neural net learns?



Preliminary Evaluation
---

Clarity - The presentation is quite clear.
Quality - The claims are not quite well enough supported. The experiments that were run don't support a clear conclusion and more experiments should have been run to support a more general conclusion.
Novelty - I don't think anyone has catalogued the performance of disentanglement methods in terms of a generalization taxonomy.
Significance - This paper might help clarify the disentanglement literature and more broadly help people think about combinatorial generalization.

I like this paper because of its clarity, novelty, and significance. However, I think the quality concerns are significant enough that it shouldn't be accepted at this stage.

Final Evaluation (Post Rebuttal)
---
The author response and accompanying paper revision clearly and effectively addressed each of the 3 main weaknesses I pointed out, so I raised my rating.

---

> ### Author Response · Authors · 2020-11-24
> **Response to AnonReviewer2**
>
> We are pleased to hear that the reviewer found the question posed in this manuscript interesting and liked our approach in investigating the issue. We have responded to the reviewer’s point 3 above in the general comments – like this reviewer, other reviewers also felt that our study needed more experiments. Consequently, we have tested these results on two more models and another dataset. Our response to the other two major concerns of the reviewer is below:
>
>
>
> Response to point 1: We agree that visualising the relationship between disentanglement and generalisation needed a bit more work. Therefore, we have now replaced Table 1 with Figure 3, which plots the relation between D-Scores and NLL for various models and conditions. If generalisation improves as a result of disentanglement, NLL should decrease as disentanglement increases. This is clearly not the case for any of our test conditions. The only exception to this is the NLL score for the perfectly disentangled (decoder) model in the Recombination-to-Range condition. But even here, the lower NLL value is misleading as the models reconstruct crucial elements of the image (the combination of left-out generative factors) incorrectly. This can be seen by looking at the example reconstructions in Figures 2 and 4 in the main text and in the Appendix, Figures 8 and 10.
>
> Regarding the results in Table 2 (now Table 1), we think that the reviewer has misunderstood the results here. It only makes sense to compare the relationship between D-scores and NLL *within* a condition. The results in Table 2 (now Table 1) only present one D-score/NLL per condition. The reason why we include only a single experiment per condition is that we found that the models trained on the image composition task learned highly disentangled representations even with beta=1. In fact, using high beta values on the full dataset actually prevented the model from learning at all. Thus, we did not vary the level of beta in order to manipulate the level of disentanglement within each generalisation condition (as we did in image reconstruction experiments). The key result here is that, despite the high D-score, models failed in their reconstruction on the critical combinations in the Recombination-to-range and Extrapolation condition.
>
>
>
> Response to point 2: Thank you – this is really useful feedback. We have now revised the section describing the taxonomy of generalization (pg. 2–3). We have removed the “Interpolation” condition as this is indeed logically equivalent to Recombination-to-Element condition – both exclude one combination.  The reviewer is also correct about Figure 1 - it only illustrates the test conditions for the three-dimensional case. To show how this taxonomy generalises to higher dimensions (more than three generative factors) we now discuss each type of test using a vector notation where it is easy to see how Extrapolation and Recombination-to-Range are a lot more challenging than the Recombination-to-Element condition. The difference between different test conditions is the number of combinations (values and generative factors) that have been excluded. This notation also makes it clear that this taxonomy can indeed be generalised to the continuous case. Throughout the manuscript, we illustrate these different conditions with specific examples of generative factors and values that were excluded from the training set to test the models.

---

### Official Review · AnonReviewer3 · 2020-11-01
**interesting, but limited study on the ability of disentanglement to generalize**

**Rating:** 5
**Confidence:** 4

**Review:**

Summary

Learning disentangled representation is often considered an important step to achieve human-like generalization. This paper studies how the degree of disentanglement affects various forms of generalization.  Variational autoencoders (VAEs) is trained with different levels of disentanglement on an unsupervised task by excluding combinations of generative factors during training. At test time the models are used to reconstruct the missing combinations in order to measure generalization performance. The paper shows that the models support only weak combinatorial generalization. The paper also tests the models in a more complex task which explicitly required independent generative factors to be controlled. The paper concludes that learning disentanglement representation is not sufficient for supporting more difficult forms of generalization.

Strengths

The paper studies 4 types of generalization, interpolation, recombination to element, Recombination to range, extrapolation.

It shows beta-VAE can achieve reasonable generalization by interpolation, not the other three types.

Weaknesses

The paper's study is limited to beta-VAE and dSprites dataset. However, it makes broad claims on the role of disentanglement in generalization.

Beta-VAE has limitations in disentanglement. It is not clear other disentanglement approaches such as Wasserstein auto-encoder, InfoGAN-CR (ICML'20) would not generalize much better.

The study is on unsupervised disentanglement. Unsupervised disentanglement has inherent limitations, see "Challenging Common Assumptions in the Unsupervised Learning of Disentangled Representations" ICML'19.

The paper should conduct experimental studies on other datasets, e.g. those in the above reference.

For image composition tasks, it states "concatenating input representations with the actions and linearly combining the resultant vectors". It will be great to explain the insights.

Decision

The paper has some interesting results on the role of disentanglement in generalization. However, the paper's study is very limited to specific model and a single dataset. Therefore, it is below acceptance threshold.

----Post-revision update---

The authors have provided results on a second dataset 3DShapes and two more models – Factor-VAE and a perfectly disentangled model. However, the construction results of the GT decoder is much worse that other models, see Figure 2; it does not reconstruct the details of the "heart" shape even for training and the edges of the "square" are not straight. This begs the question how good the GT decoder is. The open question is, what is the generalization capability of a GT decoder that can both reconstruct and disentangle perfectly?

Wasserstein auto-encoder has been shown to disentangle better and the regularization term is on the aggregate posterior instead of individual samples. Without results on WAE, the paper should refrain from making broad claims on disentanglement.  Furthermore, it would be interesting to investigate GAN based approach such as InfoGAN-CR as well.

For an experimentation paper, it should be more thorough and go beyond just two shape datasets.

I applaud the additional results the authors provided. I still think the paper is borderline (more toward 6 now). If it fixes the aforementioned weaknesses, I would recommend accept.

---

> ### Author Response · Authors · 2020-11-24
> **Response to AnonReviewer3**
>
>
>
> Thank you – this is really useful feedback. As we indicate above in our general comments, we have now replicated our results for a second dataset 3DShapes and two more models – Factor-VAE and a perfectly disentangled model. Our motivation in choosing these models and dataset has also been outlined above.
>
>
>
> Response to “The paper’s study is limited... generalise much better”: We agree with the reviewer that beta-VAEs have limitations in disentanglement. This motivated our choice of FactorVAE and the perfectly disentangled model for testing. As can be seen from Figure 3, these approaches lead to better disentanglement scores and still fail in the reconstruction of the correct combinations (Figures 2 and 4). We argue that the reason behind this is that disentanglement is not sufficient for generalisation.
>
>
>
> Response to “The study is on... ICML’19”: Our work is indeed related to that in Locatello et al. There the question is about whether it is possible to learn disentangled representations, the consistency of different approaches in learning disentanglement and whether disentanglement leads to decreased sampling complexity for downstream tasks. Our study complements theirs by asking whether disentanglement necessarily leads to better combinatorial generalisation. Performing combinatorial generalisation is a key ability of human beings and our study challenges the assumption that obtaining disentangled representations are sufficient for doing this. We have now made the connection between our approach and theirs explicit in Section 1.1, pg 2.
>
>
>
> Response to “For image composition... the insights”: Thanks – we have now provided more detail on this in Section Section 2.3, pg 8.

---

### Author Response · Authors · 2020-11-24
**Response to all reviewers:**

We would like to thank all the reviewers for the detailed and insightful comments.  The most important comment made by all the reviewers is that our strong conclusions regarding disentanglement and generalisation are limited by the fact that we only tested a small number of models using one dataset. Accordingly, our findings might be idiosyncratic to the specific experiments we carried out. We detail our response to this point here and then respond to each reviewer separately on remaining concerns.

To test the robustness of our results, we have carried out a new set of simulations that included a new dataset (3DShapes) and two new models (FactorVAE and a decoder model) in addition to the two models (VAE, beta-VAE) that we tested before.  We chose the FactorVAE model as this model has been explicitly designed to encourage independent distribution of representations, which makes it particularly well suited to understand the role of degree of disentanglement in generalisation. All models were applied to both the dSprites and 3DShapes datasets and we obtained the same pattern of results across all conditions: models fail to perform combinatorial generalization or extrapolation, except in the simplest (recombination-to-element) case. These new results are included in Section 2 and Figures 2, 3 and 4.

The new decoder model we introduce had perfect disentangled representations as we use the true latent values as inputs.  We included the decoder model in order to addresses the concern that none of the models trained end-to-end learned perfectly disentangled representations, and this raises the question as to whether there is another (untested or undiscovered) model that would learn even more disentangled representations that would generalize better.  The results with the decoder show that even perfectly disentangled representations fail in the same way.

The reviewers had also mentioned other datasets, such as 3DCars, 3DChairs and CelebA. We chose 3DShapes because this dataset explicitly lists the generative factor values for each of the images. This allowed us to selectively remove some combinations of generative factors from training to assess various forms of generalisation.

The fact that we obtained the same pattern of results in two quite different datasets for a range of different disentanglement values strengthens our conclusion that disentangled representations are not helpful for combinatorial generalisation in the models and tasks that we tested. But of course, it would be interesting to test a wider variety of datasets, models, and tasks in the future.  One of the important take-home messages of our work is that future work needs to explore new tasks and new decoding architectures in order to improve generalisation.

---

### Decision · Program_Chairs · 2021-01-07
**Final Decision**

**Decision:**

Accept (Poster)

**Comment:**

The paper seeks to empirically study and highlight how disentanglement of latent representations relates to combinatorial generalization. In particular, the main argument is to show that models fail to perform combinatorial generalization or extrapolation while succeeding in other ways. This is a borderline paper. For empirical studies it is also less agreed upon in general where one should draw the line about sufficient coverage of experiments, i.e., the burden of proof for primarily empirically derived insights. The initial submission clearly did not meet the necessary standard as the analysis was based on a single dataset and studied only two methods (VAE and beta-VAE). The revised version of the manuscript now includes additional experiments (an additional dataset and two new methods), still offering largely consistent pattern of observations, raising the paper to its current borderline status. Some questions remain about the new results (esp the decoder).